# Physiological and Metabolic Changes in Tamarillo (*Solanum betaceum*) during Fruit Ripening

**DOI:** 10.3390/molecules28041800

**Published:** 2023-02-14

**Authors:** Chaoyi Hu, Xinhao Gao, Kaiwei Dou, Changan Zhu, Yanhong Zhou, Zhangjian Hu

**Affiliations:** 1Hainan Institute, Zhejiang University, Sanya 572000, China; 2Department of Horticulture, Zhejiang University, Hangzhou 310058, China

**Keywords:** tamarillo, ethylene, fruit, metabolomics, metabolite, mature

## Abstract

Physiological and metabolic profiles in tamarillo were investigated to reveal the molecular changes during fruit maturation. The firmness, ethylene production, soluble sugar contents, and metabolomic analysis were determined in tamarillo fruit at different maturity stages. The firmness of tamarillo fruit gradually decreased during fruit ripening with increasing fructose and glucose accumulation. The rapid increase in ethylene production was found in mature fruit. Based on the untargeted metabolomic analysis, we found that amino acids, phospholipids, monosaccharides, and vitamin-related metabolites were identified as being changed during ripening. The contents of malic acid and citric acid were significantly decreased in mature fruits. Metabolites involved in phenylpropanoid biosynthesis, phenylalanine metabolism, caffeine metabolism, monoterpenoid biosynthesis, and thiamine metabolism pathways showed high abundance in mature fruits. However, we also found that most of the mature-enhanced metabolites showed reduced abundance in over-mature fruits. These results reveal the molecular profiles during tamarillo fruit maturing and suggest tamarillos have potential benefits with high nutrition and health function.

## 1. Introduction

Tamarillo (*Solanum betaceum* Cav.), also called tree tomato, is a small tree or shrub, native to the Andean region. Tamarillo is phylogenetically close to tomato (*Solanum lycopersicum* L.) and potato (*Solanum tuberosum* L.). Similar to tomato fruits, tamarillo fruits can be eaten raw, and can also be processed as dishes, sauces, juices, and other desserts. Tamarillo fruits contain high levels of carotenoids, anthocyanins, phenolic compounds, vitamins, flavonols, and dietary minerals with low caloric content [1,2,3,4,5]. Moreover, they have potential health benefits, including antioxidant, anti-obesity, anti-cancer and anti-inflammatory properties [6,7].

Fruit ripening is a complex developmental process comprising genetically programmed physiological and biochemical activities [8,9]. The texture, color, aroma, flavor, and metabolite of fruits change tremendously during the ripening process [9,10]. Usually, texture and color changes are the main observed modifications during fruit ripening. The changes in texture are caused by the modification of cell wall structure and cell turgor, and the color is changed by the decrease in chlorophyll content, with anthocyanin and/or carotenoid accumulation. Metabolites such as acids, sugars, volatiles, and others consist of flavor, aroma, and nutritional quality [11].

Fruits are usually classified into two categories based on their capacity to produce and respond to ethylene during ripening: climacteric fruits and non-climacteric fruits. There is a peak of ethylene production and respiration burst during ripening in climacteric fruits such as tomato and apple (*Malus domestica*), but not in non-climacteric fruits such as strawberry (*Fragaria vesca*) and grape (*Vitis vinifera*) [10,12,13,14]. In climacteric fruits, ethylene has long been implicated and plays a dominant role in ripening initiation [15]. Despite ethylene, the dynamic interplay between phytohormones such as auxins, abscisic acid, brassinosteroids and jasmonates, transcription factors, and epigenetic modifications also contribute to the regulation of the climacteric fruit ripening [16]. Moreover, studies indicate that climacteric (ethylene-dependent) and non-climacteric (ethylene-independent) regulation coexist during climacteric fruit ripening [17]. Ripening of the non-climacteric fruit also involves the coordinated action of different hormones, such as abscisic acid, ethylene, auxin, gibberellins, and others [11].

Liquid chromatography/mass spectrometry-based untargeted metabolomics is increasingly being used in various scenes, including plant science, food science, and disease science [18,19,20,21]. Untargeted metabolomic analysis of plant metabolites can uncover the bioactive phytochemicals in specific plant tissues and help to discover potential human health-promoting products. In this study, the physiological and molecular profiles of tamarillo and their changes during fruit ripening were determined. During the maturity of tamarillo fruits, the firmness and sugars/acid ratio was gradually decreased and enhanced, respectively, with a burst of ethylene production in mature fruits. A variety of flavor, nutrition, and antibacterial-related metabolites were identified as being changed during ripening. These data indicated tamarillo fruits as a potential functional food with health benefits.

## 2. Results

### 2.1. Changes of Physical Parameters and Ethylene Contents during Fruit Ripening

During fruit ripening, a lot of physical and molecular changes occur, including color, firmness, and phytohormones. We first observed the changes in color, firmness, and ethylene production during tamarillo fruit ripening. We found that the color of fruits was gradually turning to red from the stage of Green ripe (GR) to Break (BR), then Turning (TU), and at last, Mature (MA) in both pericarp and flesh (Figure 1A). In addition, the firmness of fruits gradually decreased from GR to MA. To test changes in ethylene production during tamarillo fruit ripening, the content of ethylene was determined in the different stages (GR, BR, TU, and MA) of tamarillo fruits (Figure 1B). Results showed that the ethylene production was at a low level in the GR and BR stages and was increased at the TU stage. At the MA stage, the ethylene production was significantly evoked, which was 12.97-fold higher than in the TU stage (Figure 1C).

### 2.2. Changes of Soluble Sugar Contents during Fruit Ripening

Sugars make a great contribution to the flavor of ripened fruits. We further monitored the change of soluble sugar, including sucrose, fructose, and glucose, during tamarillo fruit ripening and over-maturing (OM). As indicated in Figure 2A, the content of sucrose gradually increased from GR to TU, then decreased at the MA stage. Furthermore, there was no significant difference in the sucrose content between MA and OM fruits (Figure 2A). The contents of fructose and glucose showed a similar change pattern, which was both dramatically elevated after GR and peaked at MA. Fructose and glucose contents in MA fruits were 4.90-fold and 7.31-fold higher than in MG fruits, respectively. However, in OM fruits, the contents of fructose and glucose decreased slightly but not significantly (Figure 2B,C). These results showed that mature tamarillo fruits accumulate high levels of fructose and glucose with decreased sucrose content.

### 2.3. Changes of Metabolites during Fruit Ripening

A variety of metabolites are changed during fruit ripening. To obtain information on the global changes of metabolites during tamarillo fruit ripening, an untargeted metabolomic analysis was carried out in fruits at GR, BR, TU, MA, and OM stages. A total of 1707 metabolites were identified at these stages (Appendix A). Most metabolites (1512) were common in the 5 ripening stages (Figure 3A). We found that 25 metabolites uniquely existed in the MA stage, of which 14 known metabolites were annotated (Table 1). These 14 metabolites included rosmarinate, daidzin, quercetin-3-O-sophoroside, etc. Rosmarinic acid (RA) is a classical type of polyphenol that exhibits great potentially beneficial pharmacological functions such as anticarcinogenic, antiallergic, anti-inflammatory, and antimicrobial [22]. Daidzin is one of the major isoflavone glycosides and is an antioxidant isoflavonoid, which shows decreased blood alcohol levels, liver protection, and decreased inflammation activity [23,24,25]. Quercetin-3-O-sophoroside is an important glucose-bound derivative of quercetin with a dopaminergic neuroprotection effect and strong antioxidant capacity [26,27].

The principal component analysis (PCA) of both positive and negative metabolites revealed that the samples were divided into five groups, each associated with a specific fruit mature stage (Figure 3B). Among metabolites with significant differences between different stages, the contents of L-glutamic acid and D-glucose were increased following the fruit ripening (Figure 4A). The change in D-glucose content was similar to the data in Figure 2C. The contents of malic acid and citric acid were significantly decreased in MA than in GR (Figure 4A).

Moreover, 661 changed metabolites (484 upregulated and 177 downregulated) were identified in the mature fruits compared with the GR fruits (Appendix A). The Kyoto encyclopedia of genes and genomes (KEGG) compound classification indicated that these metabolites were classified into several categories, such as amino acids, phospholipids, monosaccharides, and vitamins (Figure 4B). The KEGG analysis of 661 metabolites showed that glucosinolate biosynthesis, phenylpropanoid biosynthesis, phenylalanine metabolism, caffeine metabolism, monoterpenoid biosynthesis, and thiamine metabolism pathways were significantly enriched (Figure 4C). The relative abundance of metabolites involved in these pathways was highest in MA fruits, except those involved in the glucosinolate biosynthesis pathway including 3-methyl-2-oxovaleric acid, L-tryptophane, L-tyrosine, L-tryptophan, L-phenylalanine, and L-isoleucine (Figure 4D, Appendix A).

### 2.4. Changes of Metabolites during Fruit Over-Maturing

Over-maturing may damage fruit quality. To know the effect of over-mature on tamarillo fruits, we further compared the changes of metabolites between OM and MA fruits in our metabolomic analysis data. We found that 596 metabolites were significantly changed in OM fruits than in MA fruits, of which 452 were downregulated and 144 were upregulated (Appendix A). The KEGG compound classification showed that most of these metabolites were classified as phospholipids, amino acids, carboxylic acids, steroid hormones, monosaccharides, and vitamins (Figure 5A). The KEGG analysis of these 360 metabolites showed that arginine and proline metabolism and stilbenoid, diarylheptanoid, and gingerol biosynthesis pathways were enriched (Figure 5B). Surprisingly, most of these OM-changed metabolites (360 out of 596) were common in the 661 MA-changed metabolites (significantly changed between MA and GR) (Figure 5C). Furthermore, most of these 360 metabolites showed the highest abundance specifically at the MA stage, compared to GR, BR, TU, and OM (Figure 5D), indicating the loss of the fruit quality after over-maturing.

## 3. Discussion

Ethylene plays essential roles in climacteric and non-climacteric fruits during ripening [14,15]. We found that ethylene production was significantly elevated in fruits at MA compared to previous stages (Figure 1C). This result was consistent with an earlier study, which suggested that tamarillo fruits were non-climacteric, and only traces of ethylene were produced until the final senescence in harvested mature fruits [28]. However, it was distinguished from the typical climacteric tomato fruits, in which ethylene production was soon induced after GR [29]. Previous studies in grapes reported that ethylene plays a major role in the fruit ripening and treatment of fruits with 1-methyl cyclopropane (1-MCP), a specific inhibitor of ethylene receptors, reduced ripening-related parameters [14]. In raspberry (*Rubus idaeus*) fruits, ethylene production was shown to be inversely related to firmness loss. A 1-MCP treatment delayed the firmness loss in raspberry fruits [30]. The inverse correlation between ethylene production and firmness was also shown in our data (Figure 1B,C). Future work on the relationship of MA-induced ethylene production with ripening-related parameters, such as firmness in tamarillo, can broaden our understanding of the tamarillo fruit ripening process.

The contents and equilibrium between organic acids and sugars are essential for the taste of fruits, which is vital for the choice of consumers [31]. Organic acids, citric and malic acids, mainly contribute to fleshy fruit acidity in most ripe fruits [32]. Malic acid is the predominant organic acid in apple, pear, and loquat fruits [33,34,35], while citric acid is the major organic acid in citrus [36]. Both citric and malic acids show high abundances in tamarillos [37,38]. In our metabolomic data, the content of citric acid was higher than malic acid. Furthermore, the contents of citric and malic acids were decreased in mature fruits (Figure 4A). The sugar/acid ratio is one of the most important indicators reflecting the desirable taste and sweetness of the fruits. In our study, great amounts of fructose and glucose accumulation were found in the ripening of tamarillo fruits (Figure 2B,C). This increased sugar/acid ratio may make the mature tamarillos sweeter. These results are similar to other studies of different fruits. In tomato fruits, contents of glucose and fructose were gradually elevated following maturing with a decreased level of malic acid [29]. Fructose and sucrose were increased whereas citric acid was decreased during guava (*Psidium guajava*) fruit ripening [39].

Our untargeted metabolomic analysis revealed that a variety of metabolites were changed in mature tamarillo fruits. These include essential amino acids such as L-glutamic acid, lipids, carbohydrates, and vitamins. Glucosinolate biosynthesis, phenylpropanoid biosynthesis, phenylalanine metabolism, caffeine metabolism, monoterpenoid biosynthesis, and thiamine metabolism pathways were significantly enriched during tamarillos maturing (Figure 4C). It is reported that fruit volatile compounds are mainly comprised of esters, alcohols, aldehydes, ketones, lactones, terpenoids, and apocarotenoids, and lipids, amino acids, terpenoids, and carotenoids pathways are involved in the volatile biosynthesis [40]. The phenylpropanoid pathway synthesizes flavonoids, which contribute to the red or purple color of fruits [41,42]. Phenylalanine has been shown to induce fruit’s defense response and tolerance to fungal pathogens in mango, avocado, and citrus fruit [43]. Collectively, our metabolomic analysis showed that ripening changed metabolites might confer flavor, red color, sweetness, and antibacterial activity.

RA-containing herbal preparations and food supplements are marketed with claims for health-beneficial effects, and RA is regarded as a natural antioxidant in the food industry with wide application [44]. Notably, in this study, we found that rosmarinate specifically existed in mature tamarillo fruits, suggesting the profound benefits for the health of tamarillo fruits and potential application in functional food. Moreover, a previous study showed that postharvest RA treatment delayed the ripening of tomato fruits. Exogenous application of RA decreased ethylene production, inhibited the fruit color change, but strengthened the antioxidant system by increasing both the activity of antioxidant enzymes and the contents of reduced forms of antioxidants [29]. So will the rosmarinate in mature tamarillo fruits prevent the over-maturing of fruits? Future studies about this question may provide evidence for the potential role of rosmarinate in extending tamarillo fruits’ shelf-life.

Fleshy fruit undergoes a series of development stages, an irreversible process of ripening, and eventual senescence. Senescence is the last step of the fruit life cycle and directly affects fruit quality and resistance. Advanced or over-ripening and senescence render fruit susceptible to invasion by pathogens [45]. This study found that over-mature of tamarillo fruits resulted in decreases in most metabolites compared to MA fruits. Most of the MA-induced metabolites were attenuated in OM fruits (Figure 5D), suggesting that the lower abundance of the metabolites may account for the loss of fruit quality. Phytohormone salicylic acid plays a dominant role in plant immunity, including fruit defense against pathogens. Exogenous SA treatment induces disease resistance of postharvest fruits in various species by activating H_2_O_2_ signaling or disease resistance-related genes and enzymes [46,47,48]. Our metabolic data showed that the abundance of SA was highest in MA fruits and decreased in OM fruits (Figure 4D), which may make tamarillo fruits more specious to disease when over-matured. Moreover, SA treatment has been shown to delay the ripening of bananas (*Musa acuminata*) and improve the storage life of sugar apples (*Annona squamosa*) [49,50]. It is of great significance to investigate the role of SA in extending tamarillo fruit storage time and enhancing resistance against disease.

## 4. Materials and Methods

### 4.1. Plant Materials

Tamarillo (*Solanum betaceum* Cav.) fruits of five stages (Green ripe, GR; Break, BR; Turning, TU; Mature, MA; Over-mature, OM) were harvested in Dehong, Yunnan Province (China). Fruits were selected based on mature condition according to visual color, texture, shape, and size: GR (bright green, hard texture and full size), BR (hard texture and a little color change), TU (partly vine-ripened, firm texture and light yellow color), MA (fully vine-ripened, soft texture and orange color), and OM (very soft texture). Harvested fruits were placed inside 10 L expandable polystyrene chambers with a 1 cm thick paper towel at the bottom to protect the fruit from damage and transported to the laboratory.

### 4.2. Fruit Firmness Measurement

The firmness of the tamarillo fruits pericarp was tested by the TA-XT2i texture analyzer (Stable Micro Systems Ltd., Godalming, UK) with a metal probe of 5.0 mm in diameter. The penetration depth was 10 mm, and the penetration speed was set at 1 mm/s. Newton (N) is represented as the unit of force. Each fruit was measured twice on opposite sides after removing a small piece of peel. The average of the two measurements represents the firmness of each fruit. Three biological replicates were used for each group.

### 4.3. Ethylene Production Quantification

Ethylene production was determined based on the previous study [29]. Briefly, one tamarillo fruit was sealed into a 300 mL gas-tight jar at 25 °C for 1 h. Then, a 1 mL gas sample, from the headspace, was withdrawn using a syringe and injected into a gas chromatograph (6890N GC system; Agilent, Folsom, CA, USA) fitted with a Proapack-Q column. The temperatures of the injector, detector, and oven were 140 °C, 230 °C, and 100 °C, respectively. In addition, the weight of tamarillo fruits was recorded for calculating the ethylene production on the basis of per gram of fresh weight (FW). Four biological replicates were used for each group.

### 4.4. Sugar Measurement

After firmness and ethylene quantification, tamarillo fruits were ground into fine powder in liquid nitrogen. A total of 0.1 g tamarillo fruit powder was weighed into a 10 mL tube, and then 1 mL double-distilled water (ddH_2_O) was added, followed by vortexing. The homogenates were incubated at 80 °C for 30 min. After the centrifugation at 12,000× *g* for 10 min, the supernatant was added with the same volume ddH_2_O for another incubation at 80 °C for 30 min. After the centrifugation at 12,000× *g* for 10 min, 200 µL supernatant was fulfilled with 600 µL acetonitrile (HPLC grade), followed by centrifugation at 12,000× *g* for 10 min. Then, 20 µL supernatant was fulfilled with 180 µL 80% acetonitrile, and the 200 µL solution was subjected to HPLC analysis. Standard samples of sucrose (Solarbio, Beijing, China), glucose (Solarbio, Beijing, China), and fructose (Solarbio, Beijing, China) were measured to calculate the individual sugar content in fruit samples. Three biological replicates were used for each group.

### 4.5. LC-MS/MS-Based Untargeted Metabolomics Analysis

Six biological replicates were used for each group for untargeted metabolomics analysis. Each 50 mg tamarillo fruit sample powder was accurately weighed, and the metabolites were extracted using a 400 µL methanol:water (4:1, *v*/*v*) solution with 0.02 mg/mL L-2-chlorophenylalanin as internal standard. The mixture was allowed to settle at −10 °C and treated by a high throughput tissue crusher Wonbio-96c (Shanghai wanbo biotechnology Co., Ltd., Shanghai, China) at 50 Hz for 6 min, then followed by ultrasound at 40 kHz for 30 min at 5 °C. The homogenates were placed at −20 °C for 30 min to precipitate proteins. After centrifugation at 13,000× *g* at 4 °C for 15 min, the supernatants were carefully transferred to sample vials for LC-MS/MS analysis. Untargeted metabolomics analysis was carried out using a UHPLC-Q Exactive HF-X system (Thermo Fisher Scientific, Waltham, MA, USA). A measure of 2 μL of the sample was separated by an HSS T3 column (100 mm × 2.1 mm i.d., 1.8 µm) and then entered into mass spectrometry detection. The mobile phases consisted of 0.1% formic acid in water:acetonitrile (95:5, *v*/*v*) (solvent A) and 0.1% formic acid in acetonitrile:isopropanol:water (47.5:47.5:5, *v*/*v*) (solvent B). The solvent gradient changed according to the following conditions: from 0 to 3.5 min, 0% B to 24.5% B (0.4 mL/min); from 3.5 to 5 min, 24.5% B to 65% B (0.4 mL/min); from 5 to 5.5 min, 65% B to 100% B (0.4 mL/min); from 5.5 to 7.4 min, 100% B to 100% B (0.4 mL/min to 0.6 mL/min); from 7.4 to 7.6 min, 100% B to 51.5% B (0.6 mL/min); from 7.6 to 7.8 min, 51.5% B to 0% B (0.6 mL/min to 0.5 mL/min); from 7.8 to 9 min, 0% B to 0% B (0.5 mL/min to 0.4 mL/min); and from 9 to 10 min, 0% B to 0% B (0.4 mL/min) for equilibrating the systems. The sample injection volume was 2 µL and the flow rate was set to 0.4 mL/min. The column temperature was maintained at 40 °C. During the period of analysis, all these samples were stored at 4 °C.

The mass spectrometric data were collected using a Thermo UHPLC-Q Exactive HF-X Mass Spectrometer (Thermo, Waltham, MA, USA) equipped with an electrospray ionization (ESI) source operating in either positive or negative ion mode. The optimal conditions were set as follows: heater temperature, 425 °C; capillary temperature, 325 °C; sheath gas flow rate, 50 arb; aux gas flow rate, 13 arb; ion-spray voltage floating (ISVF), −3500 V in negative mode and 3500 V in positive mode, respectively; and normalized collision energy, 20–40–60 V rolling for MS/MS. Full MS resolution was 60,000, and MS/MS resolution was 7500. Data acquisition was performed with the Data Dependent Acquisition (DDA) mode. The detection was carried out over a mass range of 70–1050 *m*/*z*.

After the mass spectrometry detection, the raw data of LC/MS were preprocessed by Progenesis QI 2.1 (Waters Corporation, Milford, CT, USA) software, and a three-dimensional data matrix in CSV format was exported. Internal standard peaks, as well as any known false positive peaks (including noise, column bleed, and derivatized reagent peaks), were removed from the data matrix, deredundant, and peak pooled. The metabolites were searched and identified in the database, and the main databases were the HMDB (http://www.hmdb.ca/ (accessed on 11 November 2022)), Metlin (https://metlin.scripps.edu/ (accessed on 11 November 2022)), and Majorbio Database. The data, after the database search, were then uploaded to the Majorbio cloud platform (https://cloud.majorbio.com (accessed on 11 November 2022)) for data analysis. Metabolic features detected at least 80% in any set of samples were retained. After filtering, minimum metabolite values were imputed for specific samples in which the metabolite levels fell below the lower limit of quantitation and each metabolic feature was normalized by sum. In order to reduce the errors caused by sample preparation and instrument instability, the response intensity of the sample mass spectrum peaks was normalized by the sum normalization method, and the normalized data matrix was obtained. The principal component analysis (PCA) and orthogonal least partial squares discriminant analysis (OPLS-DA) was performed using the R package ropls (Version 1.6.2, http://bioconductor.org/packages/release/bioc/html/ropls.html (accessed on 12 November 2022)). The selection of significantly different metabolites was determined based on the variable importance in the projection (VIP) obtained by the OPLS-DA model and the *p*-value of the Student’s *t*-test, and the metabolites with VIP > 1, *p* < 0.05 were significantly different metabolites. Differential metabolites among two groups were summarized and mapped into their biochemical pathways through metabolic enrichment and pathway analysis based on a database search (KEGG, http://www.genome.jp/kegg/ (accessed on 15 November 2022)).

### 4.6. Statistical Analysis

Experiments were conducted using a completely randomized design, all with at least three replications. Data are expressed as mean ± SD and were analyzed using GraphPad InStat software 8.3.0 for Windows (GraphPad Software Inc., La Jolla, CA, USA). Data were analyzed using one-way ANOVA tests followed by Tukey’s test. Different letters indicate significant differences between different groups.

## Figures and Tables

**Figure 1 molecules-28-01800-f001:**
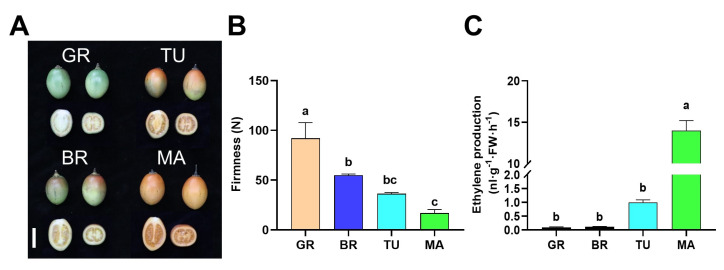
Changes of firmness and ethylene production during tamarillo fruit ripening. The phenotype (**A**), firmness (**B**), and ethylene production (**C**) of fruits at different ripening stages. Bar = 5 cm. For (**B**), data represent the mean ± SD (*n* = 3). For (**C**), data represent the mean ± SD (*n* = 4). Means denoted by the different letters significantly differ at *p* < 0.05 according to Tukey’s test. GR, Green ripe; BR, Break; TU, Turning; MA, Mature. FW: fresh weight.

**Figure 2 molecules-28-01800-f002:**
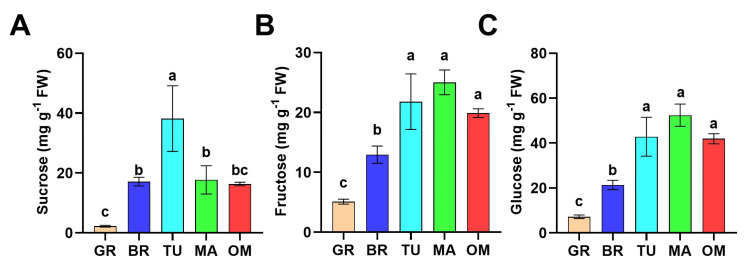
Changes of sugar accumulation during tamarillo fruit ripening. The contents of sucrose (**A**), fructose (**B**), and glucose (**C**) of fruits at different ripening stages. Data represent the mean ± SD (*n* = 3). Means denoted by the different letters significantly differ at *p* < 0.05 according to Tukey’s test. GR, Green ripe; BR, Break; TU, Turning; MA, Mature; OM, Over-mature. FW: fresh weight.

**Figure 3 molecules-28-01800-f003:**
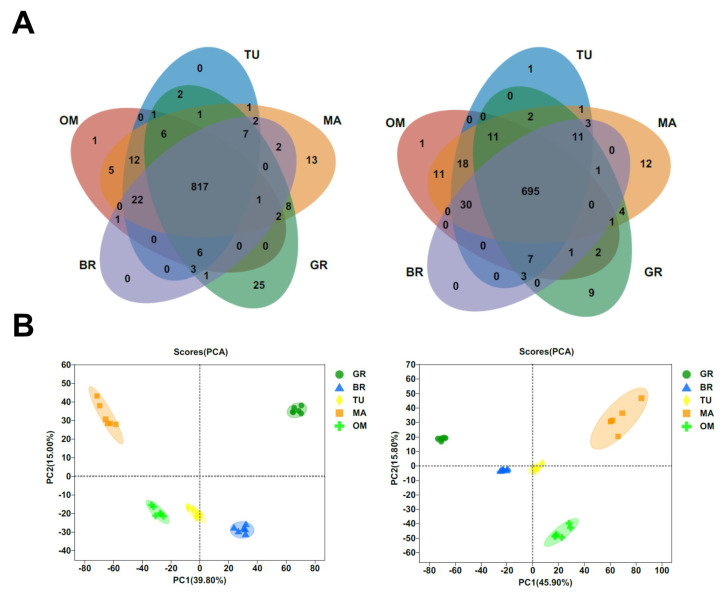
Metabolites number (**A**) and PCA of metabolites (**B**) in different ripening stages. In (**A**) and (**B**), the left and right panels represent positive and negative metabolites, respectively. GR, Green ripe; BR, Break; TU, Turning; MA, Mature; OM, Over-mature.

**Figure 4 molecules-28-01800-f004:**
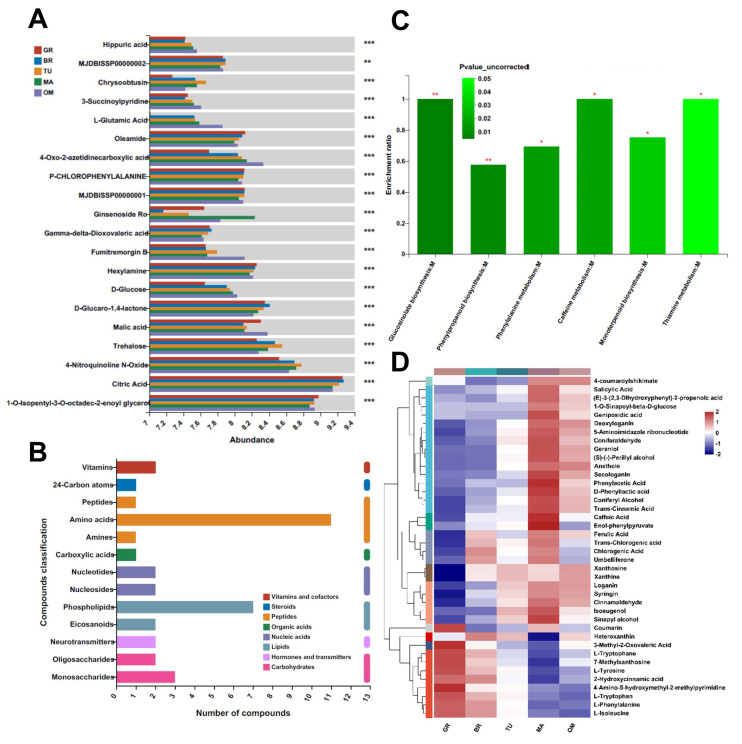
Metabolite changes during fruit ripening. Representative metabolites with significant differences between different stages (**A**). KEGG compound classification ** *p* < 0.01 and *** *p* < 0.001 according to one-way ANOVA; (**B**) and KEGG analysis (**C**) of mature changed metabolites. Heatmap indicates the relative abundance of metabolites within the enriched KEGG pathways of glucosinolate biosynthesis, phenylpropanoid biosynthesis, phenylalanine metabolism, caffeine metabolism, monoterpenoid biosynthesis, and thiamine metabolism, * *p* < 0.05 and ** *p* < 0.01 (**D**). GR, Green ripe; BR, Break; TU, Turning; MA, Mature; OM, Over-mature.

**Figure 5 molecules-28-01800-f005:**
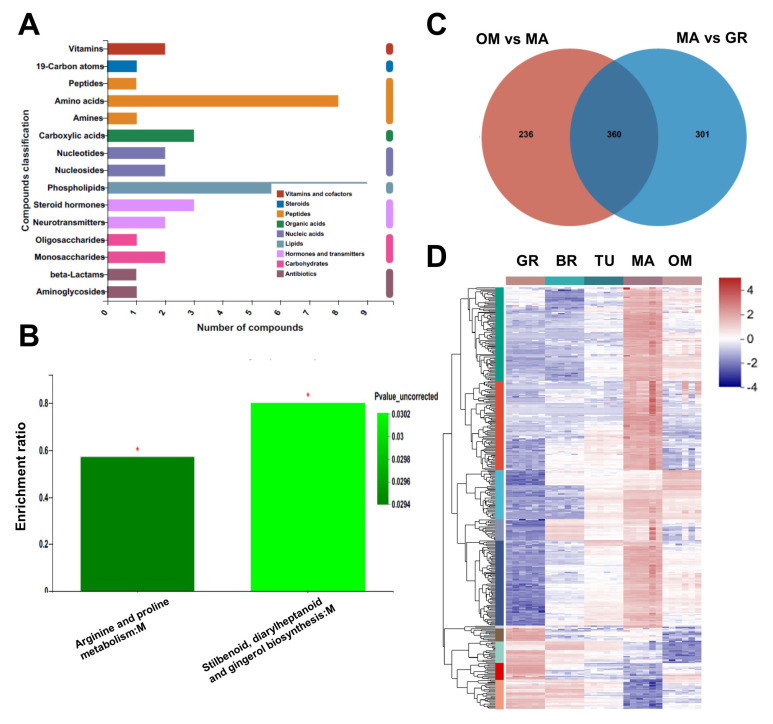
Metabolites changes during fruit over-maturing. KEGG compound classification (**A**) and KEGG analysis (**B**) of OM-changed metabolites. Venn diagram of the intersection of MA-changed and OM-changed metabolites, * *p* < 0.05; (**C**). The relative abundance of metabolites in the intersection of MA-changed and OM-changed metabolites (**D**). The original abundance was subjected to data adjustment by normalization using an R package. GR, Green ripe; BR, Break; TU, Turning; MA, Mature; OM, Over-mature.

**Table 1 molecules-28-01800-t001:** Metabolites uniquely existed in the MA stage fruits.

Metab ID	Metabolite	KEGG Pathway Description	*m*/*z*	Formula
metab_267	3,4-Di-O-caffeoylquinic acid	-	517.1343	C25H24O12
metab_333	Rosmarinate	Metabolic pathways; tyrosine metabolism	743.1585	C18H16O8
metab_871	9-Carboxymethoxymethylguanine	-	542.1426	C8H9N5O4
metab_1352	Daidzin	Biosynthesis	458.1446	C21H20O9
metab_3528	Aesculetin	-	211.0603	C9H6O4
metab_3544	Miserotoxin	-	552.2231	C9H17NO8
metab_3603	5′-N-Ethylcarboxamidoadenosine	-	341.1571	C12H16N6O4
metab_3736	Pyochelin	Biosynthesis of secondary metabolites	357.0968	C14H16N2O3S2
metab_6592	Ruberythric acid	-	515.1212	C25H26O13
metab_6846	B-D-Glucuronopyranosyl-(1->3)-a-D-galacturonopyranosyl-(1->2)-L-rhamnose	-	561.1264	C18H28O17
metab_8508	Gomphrenin II	-	717.148	C33H32N2O15
metab_8643	8-Methoxykynurenate	Tryptophan metabolism	656.1484	C11H9NO4
metab_9469	Austdiol	-	281.0672	C12H12O5
metab_10954	Quercetin-3-O-sophoroside	-	289.0836	C14H16N2O6

## Data Availability

The data presented in this study are available on request from the corresponding author.

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
