# Peer review of "Physiological and Metabolic Changes in Tamarillo (Solanum betaceum) during Fruit Ripening"

_molecules, 2023, doi:10.3390/molecules28041800_

Round 1
Reviewer 1 Report
Dear Authors
Please find my review comments below
1) For the metabolite identification, have your verified your data with other multi-omics bioinformatic platforms other than Majorbio Cloud to get confirmation?
2) Additional LC-MS method details would help reviewers and readers to help understand the section better. Please provide.
3) Graphical representation of fig-5D needs attention.
Thanks
Author Response
Thank you for your careful and patient review of our manuscript. The changes that we have made are described below:
1) For the metabolite identification, have your verified your data with other multi-omics bioinformatic platforms other than Majorbio Cloud to get confirmation?
Authors: The metabolites were searched and identified in the database, and the main database was the HMDB (http://www.hmdb.ca/), Metlin (https://metlin.scripps.edu/) and Majorbio Database. The data after the database search was then uploaded to the Majorbio cloud platform (https://cloud.majorbio.com) for data analysis.
2) Additional LC-MS method details would help reviewers and readers to help understand the section better. Please provide.
Authors: We have provided these in the revised manuscript.
3) Graphical representation of fig-5D needs attention.
Authors: Thanks for your suggestion. We have improved it.
Reviewer 2 Report
In this paper, the authors used metabolomics to analyze the metabolic changes during tamarillo fruit ripening. They found that amino acids, phospholipids, monosaccharides, vitamin-related metabolites and malic acid and citric acid changed. The results laid a foundation for further analyzing the mechanism of tamarillo fruit ripening. The article is well written, but some issues are not clarified.
1. The authors did not indicate how many biological replicates were set up in the metabolomics experiment.
2. The figure legend lacks the definition of the Error bar and how to set the data statistics for biological repetitions.
3. How to define GR, BR, TU, and MA? For example how many days after full flowering?
4. Line89-95 and Figure2, MG should be changed to GR.
5. Line92, Ma should be changed to MA.
6. There was a peak of ethylene release in the postharvest storage period (post-ripening process) of climacteric fruits, such as apple, tomato. However, this article did not give data on the change of ethylene during the post-ripening process of tamarillo fruit, so it is difficult to say that tamarillo is a climacteric fruit.
Author Response
Many thanks for your valuable suggestions and comments, which assisted us in improving our manuscript. Our response to your comments has been listed below point-by-point:
- The authors did not indicate how many biological replicates were set up in the metabolomics experiment.
Authors: Six biological replicates were used for each group for untargeted metabolomics analysis. We have added it in the revised manuscript.
- The figure legend lacks the definition of the Error bar and how to set the data statistics for biological repetitions.
Authors: We have added it.
- How to define GR, BR, TU, and MA? For example how many days after full flowering?
Authors: Fruits were selected based on mature condition according to visual color, texture, shape and size: GR (bright green, hard texture and full size), BR (hard texture and a little color changed), TU (partly vine ripened, firm texture and a light yellow color), MA (fully vine ripened, soft texture and a orange color), and OM (very soft texture). And wWe have added it in the Materials and Methods part.
- Line89-95 and Figure2, MG should be changed to GR.
Authors: We have changed it.
- Line92, Ma should be changed to MA.
Authors: We have changed it.
- There was a peak of ethylene release in the postharvest storage period (post-ripening process) of climacteric fruits, such as apple, tomato. However, this article did not give data on the change of ethylene during the post-ripening process of tamarillo fruit, so it is difficult to say that tamarillo is a climacteric fruit.
Authors: Thanks for your suggestion. We have rephrased this sentence.